# Ammonia Stress Induces Transcriptional Expression Changes in the Mature Eggs of the *Acipenser baerii*

**DOI:** 10.3390/ani15213122

**Published:** 2025-10-28

**Authors:** Qian Qi, Cheng Zhang, Wenhua Wu, Qi Zhou, Chenran Lv, Xiaohui Sun, Feng Yang

**Affiliations:** 1Henan Open Laboratory of Key Subjects of Environmental and Animal Products Safety, College of Animal Science and Technology, Henan University of Science and Technology, Luoyang 471023, China; qiqian@haust.edu.cn (Q.Q.); chengzhangcrab@163.com (C.Z.); yufu_qiq@163.com (Q.Z.); lvchenran527@163.com (C.L.); 2Heilongjiang River Fisheries Research Institute, Chinese Academy of Fishery Sciences, Harbin 150076, China; wuwenhua2008@sina.com; 3Luoyang Agricultural Technology Extension Service Center, Luoyang 471000, China; lybhcb@163.com

**Keywords:** *Acipenser baerii*, mature egg, ammonia stress, transcriptome, differentially expressed genes

## Abstract

Ammonia stress has become a significant stress factor in sturgeon aquaculture. In this study, we found that exposure to different concentrations of ammonia stress affected the fourth stage of female sturgeon egg development, primarily causing rupture of the follicle membrane, adhesion of yolk granules, and damage to the surface structure. Transcriptome analysis revealed that ammonia exposure may enrich cellular components in the extracellular space, further disrupting the function of the extracellular matrix. Antioxidant gene expression revealed that antioxidant genes were significantly downregulated under ammonia stress, and gene expression of enzymes involved in steroidogenesis was suppressed. This suggests that high ammonia concentrations impair oocyte function by inducing oxidative stress and interfering with hormone synthesis. This result enables us to understand the effect of ammonia stress on sturgeon egg development from a molecular biological perspective, providing valuable insights for future sturgeon breeding and hatching efforts.

## 1. Introduction

Ammonia is the most common inorganic pollutant in aquaculture, primarily originating from the decomposition of organic matter such as animal excrement and residual feed [1] With the expansion of intensive aquaculture scale, the accumulation of feces and feed residues has led to elevated ammonia nitrogen levels, resulting in water pollution [2]. In aquatic environments, ammonia exists in water in two forms: ionized ammonia and free ammonia. These forms are interconvertible depending on environmental conditions, with toxicity mainly attributed to free ammonia [3]. In aquatic environments, the toxicity of free ammonia is 300–400 times greater than that of ionized ammonia [4,5]. Previous studies have reported that elevated ammonia concentrations can have detrimental physiological and biochemical effects on fish. These include inhibited fish growth [6], induction of oxidative stress in intestine affecting digestion and absorption [7], immune system impairment leading to tissue inflammation [8], reduced overall immunity, induction of apoptosis in tissue cells [9], decreased cellular energy levels [10], and disruption of metabolic and respiratory functions [11]. While extensive research has focused on the effects of ammonia on tissues such as the liver, gills, and intestines, relatively few studies have investigated its toxic impact on the reproductive system, particularly on mature oocytes.

In recent years, advances in high-throughput sequencing technologies have led to revolutionary breakthroughs in the field of genomics and significant innovations in transcriptome analysis. These technologies offer broader detection ranges, higher sensitivity, and more accurate quantification of gene expression [12]. They enable the effective detection of low-abundance transcripts and allow for the resolution of complex transcriptomic phenomena such as splicing variations, dynamic editing/modification, and gene fusion. As a result, the regulation of gene expression can now be studied more systematically and comprehensively, providing robust data support for transcriptome functional research [13]. The new generation of high-throughput RNA sequencing methods (RNA Seq) enables comprehensive analysis of gene expression and associated molecular pathways under specific conditions. This method is now widely used to investigate the environmental stress response of aquatic animals, such as hypoxic stress on crucian carp (*Carassius auratus*) [14], Acute stress on Grass carp (*Ctenopharyngodon idellus*) [15], Acute hypoxia and reoxygenation on yellow catfish (*Pseudobagrus ussuriensis*) [16], ammonia stress on largemouth black bass (*Marsupenaeus japonicus*) [17], Cold stress on pomfret (*Trachinotus ovatus*) [18], and salinity stress on channel catfish (*Lctalurus punctatus*) [19]. However, research on the effects of ammonia stress on oocytes, particularly mature eggs, remains limited. A previous study reported that acute high concentration ammonia stress caused ovarian damage in goldfish by inducing mature follicle atresia [20]. Additionally, it impairs the antioxidant defense system, leading to excessive accumulation of reactive oxygen species (ROS), ovarian apoptosis, and inhibition of steroidogenesis—all of which compromise reproductive performance. Despite these findings, the underlying molecular mechanisms remain poorly understood.

*Acipenser baerii,* is a valuable cold-water species in global aquaculture. Its eggs are used to produce caviar, often referred to as “black gold”, due to their exceptional nutritional and economic value [21]. Egg quality is a critical factor affecting fish reproduction and seedling production, directly impacting the reproductive capacity of fish populations and the offspring survival rate, which are important for ensuring aquaculture efficiency [22]. In this study, high-throughput RNA sequencing technology has been used to investigate the transcriptional expression changes in mature Siberian sturgeon eggs under ammonia stress. This study aimed to identify differentially expressed genes (DEGs), analyze key signaling pathways, and elucidate the molecular response mechanism of mature eggs to ammonia stress. This research provides new insights into the environmental adaptability of fish and lays a theoretical foundation for optimizing water environment management strategies for sturgeon farming, improving artificial breeding efficiency, and ensuring the quality of Siberian sturgeon germplasm resources. It holds significant scientific and practical value for promoting the sustainable development of sturgeon aquaculture.

## 2. Materials and Methods

### 2.1. Ethical Statement

The experiments were carried out in compliance with the guidelines of the committee at Henan University of Science and Technology (protocol code HAUST-024-F0831006). Anesthetic measures were also put in place to minimize fish suffering. All tissues were removed under MS222 anesthesia (45 mg/L).

### 2.2. Experimental Siberian Sturgeon

The female Siberian sturgeon used in the experiment (body length: 111–120 cm; weight: 20.2–22.5 kg) was obtained from a sturgeon breeding facility in Guxian County, Luoyang. Before the experiment, the sturgeons were acclimated in a holding pond for 7 days. During this period, water temperature was maintained at 12 ± 1 °C, dissolved oxygen was greater than 6 mg/L, and pH was maintained at 7.6–7.8. No feeding was provided during the acclimation period. Oocyte samples were collected in vivo using a live puncture method, and the gonadal development stage was assessed through visual inspection. The eggs were at the late stage IV of development, with diameters ranging from 2.6 to 2.8 mm and weights from 0.0185 to 0.0189 g.

### 2.3. Treatment of Ammonia Stress

After the acclimation period, the Siberian sturgeons were randomly allocated into 9 circular water tanks (diameter: 10 m; depth: 80 cm), with 3 fish per tank. The experimental design 3 groups: a control group (0 mg/L, C), a low concentration group (10 mg/L, T1), and a high concentration group (50 mg/L, T2). The ammonia concentration was adjusted to the designed concentration with 10 g/L ammonium chloride (NH_4_Cl) mother liquor. One-third of the water in each tank was changed every 24 h. The ammonia concentration in water was detected using the Nessler’s reagent colorimetric method every 8 h. The exposure duration was 96 h. No feeding was provided during the experiment. One-third of the water in each tank was replaced every 24 h. Ammonia concentrations were monitored daily using the Nessler’s reagent method to ensure levels remained at the set level.

### 2.4. Mature Egg Collection

At the end of the exposure period, we found that breathe of the female fish increases as well as swimming slows down. The five replicates of Siberian sturgeons in each group were anesthetized using MS-222 and were quickly punctured for egg retrieval. A portion of the eggs was frozen in liquid nitrogen and stored at −80 °C for subsequent RNA extraction. The remaining samples were fixed in 4% paraformaldehyde for histological analysis.

### 2.5. Analysis of Mature Egg Tissue Structure

After fixation, the samples underwent graded ethanol dehydration using the following concentrations and durations: 70% (2 h), 80% (2 h), 90% (1 h), 95% (1 h), and 100% (twice for 1 h each). After dehydration, the tissues were sequentially immersed in xylene I and xylene II for 30–60 min each, until they became fully transparent. For paraffin embedding, the transparent tissues were first infiltrated with a xylene-paraffin mixture (1:1) for 1 h, followed by sequential immersion in paraffin I and paraffin II for 50–60 min each to ensure complete infiltration. Melted paraffin was then poured into embedding molds. The tissues were placed at the center of the mold using tweezers, properly oriented, and rapidly solidified on a cold plate or ice surface to form embedding blocks.

Sections were cut at 3–5 μm thickness using a paraffin slicer. The sections were put in warm water at 40–45 °C to allow the sections to unfold naturally, then mounted onto glass slides and air-dried. After drying, routine staining with Hematoxylin and Eosin (H&E) was performed. Finally, the stained sections were examined and imaged using a Motic BA400 digital microscope (Xiamen, China).

### 2.6. RNA Extraction and Quality Testing

RNA extraction from mature eggs was conducted using the Trizol method [23]. RNA concentration and purity were assessed using a Nanodrop spectrophotometer, while the RNA integrity was evaluated using 1% agarose gel electrophoresis. Only RNA samples that met the quality standards were selected for library construction and transcriptome sequencing.

### 2.7. RNA Library Construction, Sequencing, and Quality Control

This study collected a total of 15 mature egg samples, and each group had 5 replicates. Qualified RNA samples were sent to Shanghai Meiji Biomedical Technology Co., Ltd (Shanghai, China). For RNA Seq library construction and sequencing. Paired-end sequencing (150 bp, PE150) was performed on the Illumina NovaSeq 6000 platform (Shanghai, China), generating no less than 6 GB of data per sample. All processes followed Illumina’s official standard operating procedures (SOP) to ensure high data quality and sequencing accuracy. After sequencing, the quality of the raw sequencing data was assessed using FastQC software (v 0.115). Clean reads were obtained using Trimomatic (v 0.39) by removing reads containing adapters, reads contaminated with adapters, reads containing all A bases, reads containing more than 10% N, and reads containing over 50% low-quality bases (Q ≤ 20).

### 2.8. Validation of Data via RT-qPCR

The 9 RNA samples (each group with bio-triplicate) were treated with gDNA eraser (PrimeScript™ RT reagent Kit with gDNA Eraser, Takara, Dalian, China) to eliminate residual genomic DNA, and 400 ng RNA in each sample was used to synthesize first-strand cDNA with PrimeScript^TM^ RT reagent Kit (Takara, Dalian, China). With the cDNA as the template, qPCR was conducted to examine the expression patterns of 5 DEGs, including *GCLM*, *GST*, *CYP11A1*, *CYP17*, *CYP19*, and *3βHSD.* The primers were designed in NCBI primer blast (Appendix A). Each reaction was carried out in a total volume of 10 μL, containing 3 μM of each primer, 1 μL of cDNA, 1 μL of H_2_O, and 5 μL of 2× SYBR Green Super mix (Takara, Dalian, China). qPCR was performed in 96-well plates on a CFX96 Real-time PCR System, following a two-temperature cycle protocol: an initial denaturation at 95 °C for 3 min was followed by 39 cycles of 95 °C denaturation for 5 s and 60 °C annealing and extension for 20 s. In the end, melting curve analysis was conducted with the temperature gradually increasing from 65 °C to 95 °C in 0.5 increments. Each sample was run in triplicate. *β -actin* gene was used as the reference to normalize the transcription level of *GCLM*, *GST*, *CYP11A1*, *CYP17*, *CYP19*, and *3βHSD* [24]. Relative changes in mRNA transcript expression were calculated using the 2^−ΔΔCT^ method, where ΔCT = CT (target gene) − CT (reference gene) and ΔΔCT = ΔCT (experimental) − ΔCT (control).

### 2.9. De Novo Assembly and Functional Annotation

Trinity was used to perform de novo assembly on all clean reads, generating optimized transcript assemblies and obtaining unigenes. Functional annotation of the unigenes was conducted by aligning them against multiple databases, including the NR, Swiss-Prot, Pfam, EggNOG, GO, and KEGG.

### 2.10. Data Processing

Quantitative analysis of gene expression levels was performed using RSEM (v1.55.0) software, and the gene expression level FPKM (fragments per kill of transcript per mill) was calculated. Differential expression analysis between groups was conducted using DESeq2 software (v1.48.1), with |log_2_ Fold change| ≥ 1 and *p* < 0.05 as the standard DEGs. GO functional enrichment and KEGG pathway enrichment analysis on DEGs were performed using Goatools software (v1.5.2) and the self-developed process of Meiji Biotechnology, respectively, identifying significantly enriched biological processes and metabolic pathways (*p* < 0.05).

### 2.11. WGCNA

Co-expression networks for all differentially expressed genes (DEGs) were constructed using Weighted Gene Co-expression Network Analysis (WGCNA) (R studio v1.41106). Sample outliers were identified and removed using the cutreeStatic function. The soft thresholding power (β) was selected as the minimum value that achieved a high scale-free topology fit index (Appendix A). The different modules were identified by a self-developed process of Meiji Biotechnology.

## 3. Results

### 3.1. Effects of Ammonia Stress on the Microstructure of Mature Eggs of Siberian Sturgeons

Histological observations of mature eggs of Siberian sturgeon after 96 h of ammonia exposure are presented in Figure 1. In the C group, egg cells contained intact yolk granules, and the follicular membranes were clear and intact. In the T1 group, the follicular membranes remained clear and intact, but larger vacuoles were observed, and vacuole fusion had begun. In the T2 group, the mature eggs had ruptured follicular membranes and adherent yolk granules.

### 3.2. Transcriptome Sequencing Result Analysis

A total of 97.89 Gb of clean data was obtained from 15 samples, with each sample yielding more than 6.09 Gb. The Q30 base percentage was more than 96.26%, and the average GC content percentage was 49.41%. (Table 1). The transcriptome data of five biological replicates from each ammonia stress exhibited a good correlation (Appendix A). The results confirmed that the sequencing data can be used for subsequent analysis. Do novo assembly using Trinity software (v2.11.0) produced 136,959 transcripts and 76,162 Unigenes, with N50 values of 2287 and 2153 for transcripts and Unigenes, respectively. The average Unigene length was 1039.50 bp, with the highest proportion (52%) ranging between 200 and 500 bp (Table 2).

### 3.3. Unigenes Function Annotations

The unigene sequence was aligned with six major databases to obtain unigene annotation information. Among the 75,197 Unigenes identified from the 15 mature egg samples, 33,334 Unigenes were successfully annotated, accounting for approximately 44.33%. Functional annotation was performed using six major databases: NR, Swiss Prot, Pfam, EggNOG, GO, and KEGG. Among these, the NR database had the highest annotation rate at 43.54% (32,742 unigenes), while the Pfam database had the lowest annotation rate at 28.12% (21,145 unigenes).

### 3.4. Differential Expression Analysis

PCA analysis indicated that samples within the same group were close to each other and exhibited good repeatability (Figure 2A). The comparison between T1 and T2 groups revealed 9446 DEGs, including 5337 upregulated and 4109 downregulated genes. These results indicated that the number of DEGs in mature Siberian sturgeon eggs decreases with increasing ammonia concentration. Plotting of three sets of DEGs demonstrated that there were 1759 unique DEGs detected between the T1 vs. C groups, 762 unique DEGs between the T2 and C groups, 4907 unique DEGs between the T2 vs. T1 groups, and 266 unique DEGs between all three groups (Figure 2B).

Additionally, compared to the C group, the T1 group yielded 5576 DEGs (Figure 2C), including 1987 upregulated and 3589 downregulated genes (Figure 2D). The T2 group showed 3719 DEGs, including 1590 upregulated and 2129 downregulated genes (Figure 2E).

### 3.5. Enrichment Analysis of DEGs

GO functional enrichment analysis was performed on DEGs and the results showed a significant enrichment in biological processes (BP), cellular components (CC), and molecular functions (MF) (Figure 3). In the T1 vs. C comparison, DEGs in the BP category were significantly enriched in multicellular organic processes. Within the CC category, DEGs were significantly enriched in extracellular spaces and other aspects, while in the MF category, DEGs were significantly enriched in signal receptor regulatory factor activity and receptor ligand activity (Figure 3A). In the T2 vs. C comparison, DEGs were significantly enriched in cell surface receptor signaling pathways within the BP category, while in the CC category, DEGs were significantly enriched in extracellular spaces and other aspects. In the MF category, DEGs were significantly enriched in signal receptor binding (Figure 3B).

In the T2 vs. T1 comparison, DEGs were significantly enriched in immune system processes within the BP category, while in the CC category, they were significantly enriched in extracellular spaces and other aspects. In the MF category, DEGs were significantly enriched in signal receptor regulatory factor activity and receptor ligand activity (Figure 3C).

The results of KEGG enrichment analysis demonstrated that the pathways enriched with DEGs in T1 vs. C group (Figure 4A) and T2 vs. C group (Figure 4B) were related to several metabolic pathways, including histidine metabolism (map00340), arginine and proline metabolism (map00330), tyrosine metabolism (map00350), alanine, aspartate and glutamate metabolism (map00250), cysteine and methionine metabolism (map00270), glycine, serine and threonine metabolism (map00260), valine, leucine and isoleucine degradation (map00280).

### 3.6. WCGNA Analysis

To identify the different co-expressed modules under ammonia stress in mature eggs of *A. baerii*, WGCNA was conducted on the 26,369 core DEGs. A total of 29 distinct co-expressed modules (labeled with various colors) were displayed on the dendrogram (Figure 4A), The modules of turquoise and blue had the largest number of DEGs (Appendix A). 14 modules had a positive correlation with the high concentrations of ammonia stress, suggesting that genes in these modules positively regulate ammonia tolerance in mature eggs of *A. baerii* (Figure 4B). Therefore, to promote tolerance under high concentrations of ammonia stress, these genes were obviously upregulated. In contrast, 11 modules were positively associated with low concentrations of ammonia stress, especially for MEred, with a correlation coefficient(r) of 0.818. This phenomenon may display possible correlations between genes that determine different ammonia stress-resistance traits.

### 3.7. RNA-Seq Expression Analysis

In addition, pathways related to antioxidant activity were also enriched, including glutathione metabolism (map00480) and chemical carcinogenic reactive oxygen species (map05208). Notably, *GCLM* gene was significantly downregulated, while *GST* gene was upregulated in T2 vs. C group (Figure 5). Furthermore, the ovarian steroidogenesis pathway (map04913), which is crucial for reproductive regulation, showed significant changes. In the T1 vs. C group, the expressions of *CYP11A1*, *3βHSD*, and *CYP19a1* were significantly downregulated (Figure 5). Similarly, in the T2 vs. C group, *CYP11A1*, *CYP17*, and *CYP19a1* were significantly downregulated (Figure 5). These findings suggest that ammonia stress may impair steroid hormone synthesis, potentially affecting reproductive function and oocyte maturation. The relative mRNA expression levels of 6 key genes selected from DEGs were validated by RT-qPCR. A comparison of the fold changes obtained from qPCR and DGE analysis revealed a consistent expression trend for all 6 genes (Figure 5). Therefore, we verified the reliability of the RNA-seq (quantification) results.

## 4. Discussion

### 4.1. Different Performance of the Mature Eggs Under the Ammonia Stress

With the ongoing expansion of aquaculture, environmental factors have become major constraints on fish farming efficiency. Among these, ammonia is a critical pollutant that significantly alters water quality [25]. When the water ammonia concentrations exceed the acceptable levels, excess ammonia gradually accumulates in various tissues of aquatic animals [26]. This accumulation causes a series of adverse reactions, such as the release of ROS, which causes oxidative stress and interferes with their normal physiological functions, resulting in functional disorders. Consequently, aquatic animals twitch and die [27]. Previous studies in aquatic animals reported that several body organs were affected due to ammonia stress, including brain [28,29], intestine [7,30], gill [30], liver [17,29], and ovary [20]. In this study, we used high-throughput RNA sequencing technology to investigate the transcriptional expression changes in mature Siberian sturgeon eggs in different ammonia concentrations. Some DEGs were identified as well as various signaling pathways were enriched under the high-concentration ammonia stress. It may change metabolism and interfere with the structure of the mature eggs of *A. baerii*. The ovary is a vital organ for the production of germ cells and plays a central role in reproduction. As a key carrier of the reproductive process, mature eggs are vulnerable to environmental changes, such as starvation, pH, light, temperature, and density [31,32]. Egg quality is influenced by yolk formation and oocyte development [32]. When mature eggs are exposed to an ammonia environment, exploring the changes in gene expression and physiological and biochemical indicators in eggs can help elucidate the molecular and physiological mechanisms of germ cells under ammonia stress. Such insights provide a theoretical foundation for understanding how ammonia affects reproductive function in aquatic animals. Although transcriptome sequencing technology has been used to study the ammonia stress responses in the ovaries of goldfish [20], the molecular effects of ammonia stress on the mature eggs of Siberian sturgeon have remained unexplored. In the present study, transcriptome analysis has been used to investigate the molecular response mechanism of Siberian sturgeon mature eggs subjected to various concentrations of ammonia.

### 4.2. Ammonia Stress Increased Leads to a Larger Number of DEGs

Based on the classification of atretic follicles by hunter distinct signs of follicular atresia such as rupture of the follicular membrane and vacuolization of the yolk granules were observed in T2 group. In contrast, the overall structure of mature eggs in the T1 group was largely intact, with no clear signs of atresia. Follicular atresia is a macroscopic manifestation of oocyte apoptosis, and its molecular mechanism may be closely related to the activation of apoptosis-related pathways. A previous study on goldfish ovaries stated that acute high-concentration ammonia stress caused a significant increase in ovarian apoptosis rate through ROS-mediated oxidative stress [20]. The results of this study align with those results, suggesting that ammonia-induced follicular damage in Siberian sturgeon may also be mediated through a conserved oxidative stress–apoptosis pathway. Therefore, minimizing environmental stress and limiting exposure duration are essential during fish management to ensure the production of high-quality mature eggs. In the present study, after 96 h of ammonia stress, 5576 and 3719 DEGs were detected in T1 vs. C group and T2 vs. C group, respectively. There was a notable decrease in the number of DEGs with the increase in ammonia concentration, suggesting that lower concentrations may trigger adaptive cellular responses, while higher concentrations disrupt intracellular balance and transcriptional regulation.

### 4.3. Significant GO and KEGG Pathways in Ammonia Stress Response

Go enrichment analysis showed that in the three comparison groups, the extracellular space was significantly enriched in the CC category, indicating potential interference of ammonia stress with the composition, structure or function of the extracellular matrix of Siberian sturgeon mature eggs, as well as the altered expression of the extracellular proteins. The extracellular space serves as the first barrier between cells and the external environment (including dissolved ammonia), which is crucial for cell–cell communication and signal transmission [33]. This enrichment suggests that sturgeon eggs may buffer ammonia toxicity by remodeling the extracellular matrix and enhancing secretion activity and providing a material basis for subsequent signaling. In the MF category, T1 vs. C group and T2 vs. T1 group were significantly enriched in the activity of signal receptor regulators and receptor ligands, while T2 vs. C group was dominated by signal receptor binding, indicating that there were hierarchical differences in the regulation of signaling pathways in eggs under different concentrations of ammonia stress (Figure 3B). Low concentration ammonia stress may enhance signaling efficiency via upregulating signal receptor regulators, while high concentration ammonia stress relies more on the direct binding ability of receptors and ligands for rapid stress response activation. This dynamic regulation may serve as a strategy for mature eggs to maintain cell communication under ammonia stress. In the BP category, T1 vs. C group was enriched in multicellular organism processes, reflecting mechanisms related to organism regulation and development under low ammonia stress. In contrast, the T2 vs. C group demonstrated enrichment in cell surface receptor signaling pathway, aligning with the molecular function findings and highlighting the prominence of signal transduction under high ammonia stress. The enriched immune system process in the T2 vs. T1 group highlighted the role of the high concentration of ammonia in stimulating the immune response of mature eggs and participating in the regulation of damage perception and repair mechanisms. When fish are subjected to ammonia stress, their bodies promote the decomposition of ammonia compounds to prevent ammonia poisoning [34]. Ammonia can be metabolized inside fish bodies by two ways. For the one way, glutamine is formed from ammonia and glutamate inside the cells, then Glutamine reaches the gill through the circulation of blood and is decomposed into glutamate and ammonia again, and finally it is excreted in the form of free ammonia. Another way is ammonia excretion as urea compounds. KEGG enrichment analysis demonstrated that ammonia stress influenced the metabolism of a variety of amino acids, including histidine, arginine, proline, tyrosine, alanine, aspartate, glutamate, cysteine, methionine, glycine, serine, and threonine. In addition, it affected valine, leucine, and isoleucine degradation. These pathways suggest that Siberian mature eggs may cope with ammonia stress by adjusting amino acid metabolism.

The normal concentration of ROS in the ovaries is conducive to ovarian development and ovulation. When the ROS concentration exceeds the acceptable levels, apoptosis of ovarian granulosa cells and oocytes occurs, resulting in premature ovarian failure, which is not conducive to ovulation [20]. Higher levels of ROS in the tissues cause an increase in the peroxidation and body damage. In this study, genes encoding ROS scavengers GST and GCLM were downregulated in the T2 group, indicating severe oxidative stress and ROS production due to high ammonia concentrations.

### 4.4. Hunt for Key Modules Through WGCNA

WGCNA aims to identify co-expressed gene modules and investigate the relationships between gene networks, as well as the core genes within the network. In this study, we used WGCNA to divide the core DEGs into 29 modules (Figure 4B). We found that MEblue (containing 3840 genes) and MEdarklurquoise (containing 3902) were positively related to high concentrations of ammonia stress, while the MEmidnight and MEred modules were positively associated with control and low concentrations of ammonia stress, with correlation coefficients of more than 0.8, respectively. However, MEpink (containing 1149 genes) and MEcyan (containing 461 genes) were negatively correlated with low and high concentrations of ammonia stress, with correlation coefficients greater than −0.8. It may indicate that some of the gene expressions in these modules were prohibited under ammonia stress. Several genes in these DEGs have been reported to be involved in the steroidogenic processing pathway. Such as *CYP17*, located in the smooth endoplasmic reticulum, converts progesterone to androgen precursor androstenedione (A4) by catalyzing 17α-hydroxylation and 17,20-cleavage of progesterone, followed by 17β-hsd catalyzed conv to testosterone (T). Therefore, *CYP17* is a vital enzyme in the biosynthetic pathway from progesterone to testosterone. Additionally, *CYP19A1*, predominantly expressed in the ovary, encodes aromatase, which converts testosterone (T) to estrogen (E_2_), playing a crucial role in gonadal development [35,36]. Steroid hormone synthesis, crucial for ovarian development was also impacted by ammonia stress [37]. Synthesis of sex hormones requires substrates such as Cholesterol. Star protein is a key cholesterol transporter involved in cholesterol metabolism [36]. When the star protein is damaged, the level of sex hormone synthesis is reduced, and interference with sex hormone synthesis pathways including soybean isoflavones occurs [38]. Various steroidogenic enzymes are essential for the biosynthesis of sex steroids [39]. Among them, *CYP11* is a member of the Cytochrome P450 family and has two subtypes: *CYP11A1* and *CYP11B1*. The *CYP11A1* gene encodes cholesterol side-chain lyase, which catalyzes the steroid conversion to pregnenolone. This reaction occurs in the inner mitochondrial membrane and represents a crucial regulatory step in steroid hormone biosynthesis. 3β- hydroxylated steroid dehydrogenase (3β HSD), located in the smooth endoplasmic reticulum, converts pregnenolone (P5) to progesterone (P4) through the dehydrogenation of C3 and the transfer of the C5 double bond to C4. This represents the second key step in steroidogenesis. In this study, *CYP11A1*, *3βHSD* and *CYP19A1* were significantly downregulated in T1 group (Figure 5, while *CYP11A1*, *CYP17* and *CYP19A1* were significantly downregulated in T2 group (Figure 5). These results suggest that ammonia exposure may interfere with the steroidogenic pathway in Siberian sturgeon, potentially disrupting normal reproductive function.

## 5. Conclusions

In this study, we found that low and short-term ammonia stress compromises the structural integrity of mature eggs of Siberian sturgeon, particularly under high ammonia concentration. GO and KEGG analyses highlighted signal receptor regulators and receptor ligands as key biological molecular functions (MF) involved in ammonia stress. Ammonia stress induces oxidative stress of antioxidant genes such as *GST* and *GCLM*, changing the metabolism of amino acids such as histidine and arginine, and downregulating genes of key steroidogenic enzymes such as *CYP11A1* and *CYP19A1*. Additionally, it influences sex hormone synthesis, which not only causes structural damage and functional impairment of the mature eggs but may also affect ovulation and fertilization processes. Consequently, ammonia stress poses a significant threat to reproductive success and seedling production in Siberian sturgeon aquaculture.

## Figures and Tables

**Figure 1 animals-15-03122-f001:**
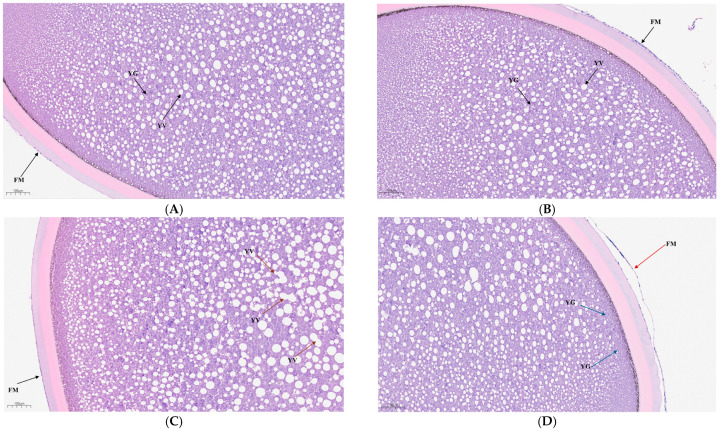
The impact of ammonia stress on the histopathology of mature eggs of *A. baerii* (96 h): Group C (**A**,**B**); Group T1 (**C**); Group T2 (**D**). FM, follicular membrane; YV, yolk vesicle; YG, yolk granule; yellow arrows indicate vacuolar fusion; blue arrows indicate the beginning of yolk granule adhesion; red arrow indicates follicular membrane rupture.

**Figure 2 animals-15-03122-f002:**
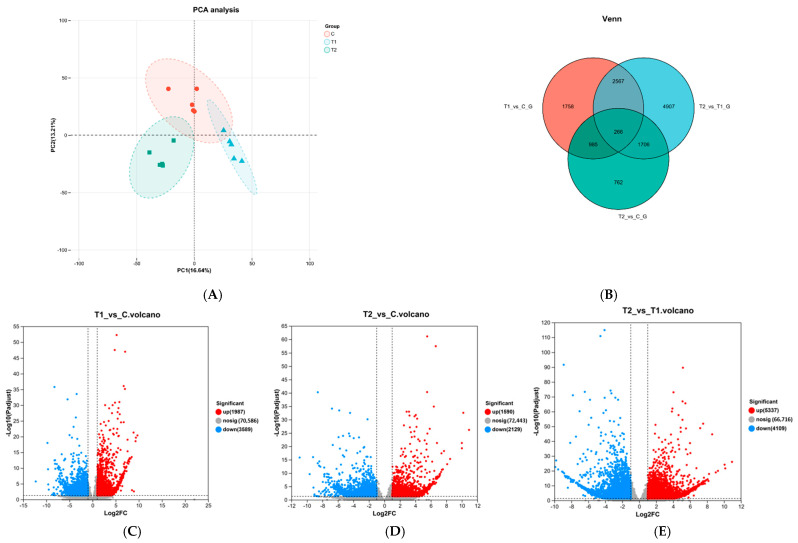
Transcriptome data analysis and differential expression gene quantity analysis: PCA analysis (**A**); Venne diagram of DEGs between three groups (**B**); Volcano diagram of significant differences in gene distribution between T1 vs. C (**C**), T2 vs. C (**D**), T1 vs. T2 (**E**).

**Figure 3 animals-15-03122-f003:**
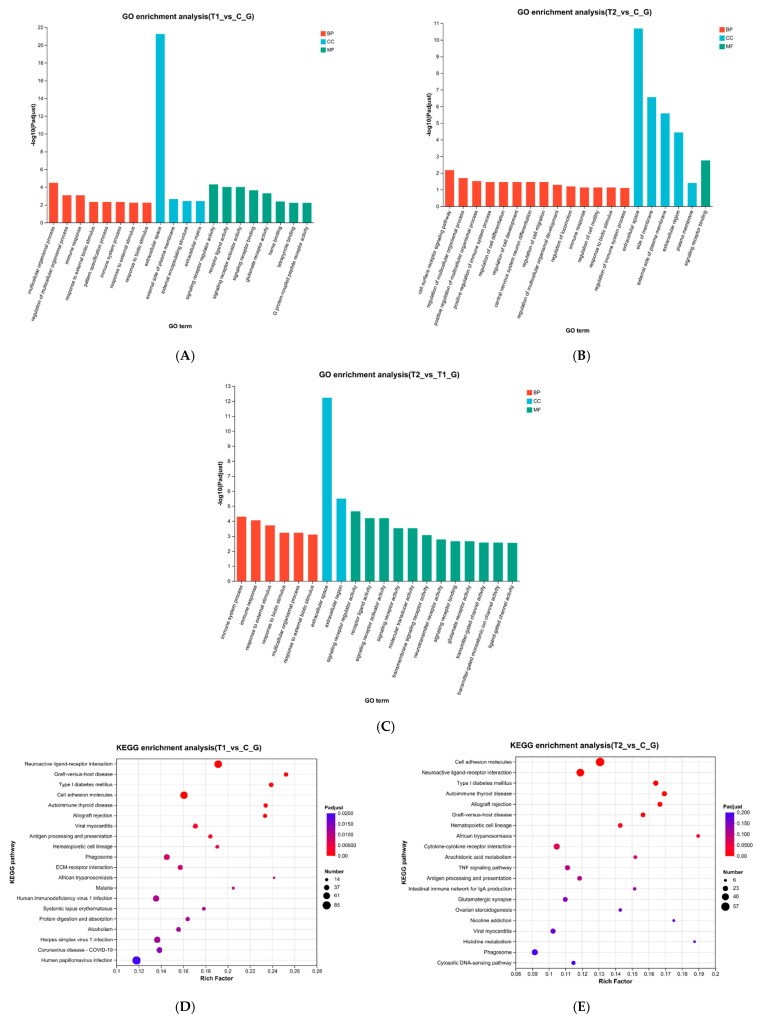
Functional enrichment analysis of DEGs: GO functional enrichment analysis of differentially expressed genes between T1 vs. C (**A**), T2 vs. C (**B**), T1 vs. T2 (**C**); KEGG enrichment analysis of differentially expressed genes T1 vs. C (**D**), T2 vs. C (**E**).

**Figure 4 animals-15-03122-f004:**
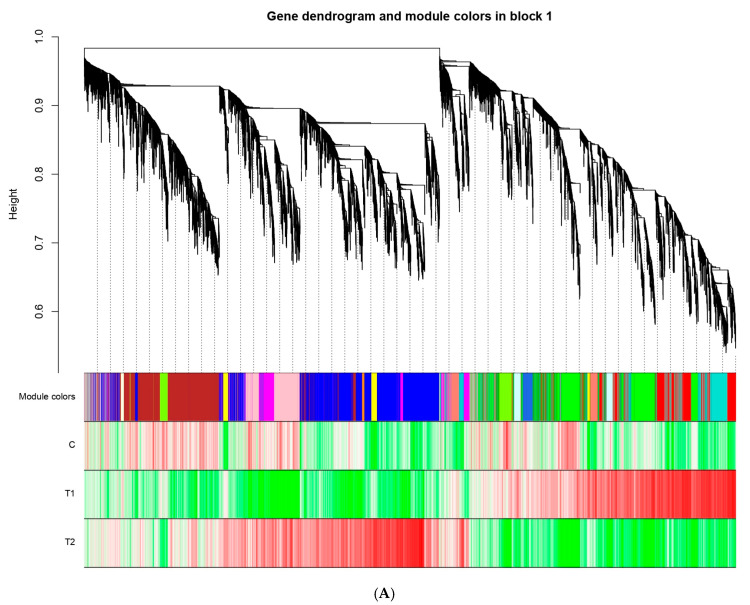
WGCNA: (**A**) Hierarchical cluster tree on the upper part showing co-expression modules identified by the Dynamic Tree Cut method. the middle part shows the modules to a short vertical line, which corresponds to a gene. Genes with highly co-expressed levels were merged into one module. Branches of the dendrogram group together densely interconnected modules and are labeled with different colors. The lower part displays a heatmap of the correlation between genes within the modules and traits. Each row represents a phenotype, and each column represents a gene within the module. The colors indicate the magnitude of the correlation, with red representing a positive correlation and green representing a negative correlation. (**B**) Module–trait associations showing Module–Trait Relationships (MTRs) of different modules under control and different concentrations of ammonia stress. The numbers represent the Pearson correlation coefficients. Positive correlation is colored in red, while negative correlation is colored in blue.

**Figure 5 animals-15-03122-f005:**
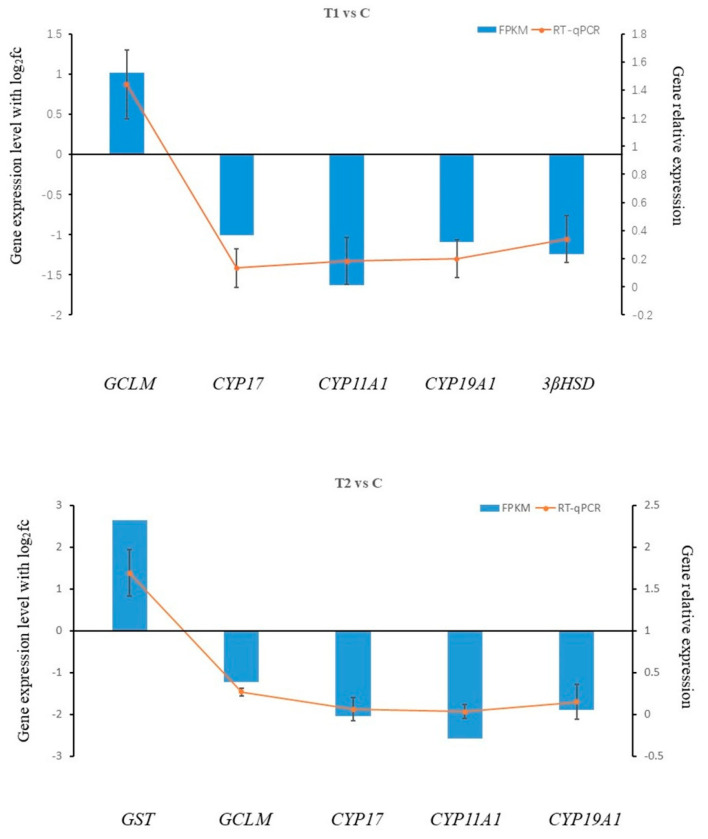
The validation of 6 genes related to DEGs in RNA-seq with RT-qPCR under the various ammonia stress conditions, relative expression level determined from five biological replicates by RT-qPCR using the 2^−ΔΔCT^ method.

**Table 1 animals-15-03122-t001:** Statistical summary results of transcriptome sequencing data.

Sample	Raw Reads	Clean Reads	Error Rate	Q20/%	Q30/%	GC/%
C1	44,177,344	43,759,336	0.0117	99.25	96.56	48.77
C2	43,025,390	42,649,676	0.0116	99.27	96.61	48.48
C3	43,511,676	43,108,596	0.0117	99.25	96.54	49.37
C4	43,393,402	42,969,304	0.0117	99.25	96.53	51.12
C5	42,964,220	42,579,062	0.0117	99.23	96.49	49.88
T1_1	42,424,566	42,089,948	0.0116	99.3	96.73	48.85
T1_2	43,138,566	42,806,416	0.0116	99.31	96.73	49.57
T1_3	43,132,158	42,848,662	0.0115	99.31	96.75	49.86
T1_4	43,624,112	43,297,704	0.0115	99.33	96.81	48.48
T1_5	43,560,080	43,244,660	0.0115	99.32	96.77	50.21
T2_1	43,536,888	43,086,176	0.0117	99.24	96.55	49.67
T2_2	42,724,446	42,363,790	0.0117	99.25	96.53	50.09
T2_3	40,971,686	40,593,188	0.0116	99.27	96.63	48.75
T2_4	46,351,226	45,775,014	0.0118	99.18	96.29	49.39
T2_5	51,754,234	51,172,172	0.0118	99.17	96.26	48.66

**Table 2 animals-15-03122-t002:** Assembly results.

Length Range/bp	Transcript	Unigene
200~500	59,670	39,270
501~1000	29,440	16,100
1001~2001	22,789	9621
>2001	25,060	11,171
Total number	136,959	76,162
N50 length	2287	2153
Mean length	1195.71	1039.50

## Data Availability

The data that support the findings in this study are available from the corresponding author upon reasonable request.

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
