# Peer review of "Ammonia Stress Induces Transcriptional Expression Changes in the Mature Eggs of the *Acipenser baerii"

_animals, 2025, doi:10.3390/ani15213122_

Round 1

Reviewer 1 Report

Comments and Suggestions for Authors

It is very good work. This work explored exposure to different concentrations of ammonia stress affected the fourth stage of female sturgeon egg development. A very novel perspective. It found high ammonia concentrations impair oocyte function by inducing oxidative stress and interfering with hormone synthesis.

Figure1, The scale is inconsistent.

Figure5 need to be modified, the image is a bit stretched.

LINE 429: Ijiri Deng et al., 2009.

The authors of reference 5 was not correct.

Whether you have detected the fertilization Rate, which has more practical value?

Author Response

Comment 1: Figure1, The scale is inconsistent.

Reply: Thank you for pointing this out, we agree with this comment. Therefore, we have modified the scale. Please check it on Page 6. Figure1

Comment 2: Figure5 need to be modified, the image is a bit stretched.

Reply: Thank you for pointing this out, we agree with this comment. Therefore, we have modified the picture.  Please check it on page 12 Figure 5

Comment 3: LINE 429: Ijiri Deng et al., 2009.

Reply: Thank you for pointing this out, we agree with this comment. Therefore, we have modified the reference. please check it on page 15 ,Line 500

Comment 4: The authors of reference 5 was not correct.

Reply: Thank you for pointing this out, we agree with this comment. Therefore, we have modified the reference.

Reviewer 2 Report

Comments and Suggestions for Authors

The present manuscript is dedicated to the study of the effect of various concentrations of ammonium on the morphology and transcriptomic profile of Siberian sturgeon eggs. Ammonium is among the most prevalent pollutants in aquatic environments, exerting a substantial toxic effect on aquatic organisms. A particular emphasis is placed on the regulation of this substance's concentration within fish farms. The augmentation of its levels during intensive feeding can result in a substantial decline in growth rate and, in extreme cases, even lead to the demise of farmed individuals.

The manuscript's strengths lie in its comprehensive account of the transcriptomic analysis' outcomes and its meticulous examination of the derived data. Specifically, the authors establish a correlation between the morphological changes observed in oocytes exposed to ammonium and atresia, as well as activation of apoptosis-related pathways. They further demonstrate a link between these changes and increased oxidative stress, as well as impaired steroid hormone synthesis. Furthermore, the study revealed that mature oocytes have the capacity to mitigate the deleterious effects of ammonium by modulating amino acid metabolism. From a pragmatic standpoint, the study is pertinent as sturgeons represent a highly valuable fish species, and caviar constitutes a pivotal product derived from them.

The work possesses both scientific and practical significance; however, the authors must supplement the "Materials and Methods" section to enhance the reproducibility of the results.

Specific comments:

- The authors should specify the amount of biomaterial (e.g., 5–10 g) collected for research (line 128).
- The “Materials and Methods” section states that during the experimental exposure, the fish were kept in pools with a diameter of 1 m (line 133). How appropriate is it to keep specimens measuring 111–120 cm (line 122) in such containers? Additionally, the authors should specify the depth of the pools.
- The authors must provide a thorough justification for the selection of ammonium concentrations employed in the study. Additionally, the researchers should specify which inorganic salt or other reagent was utilized in order to obtain the stated concentrations. It is my estimation that these concentrations approximate lethal levels; therefore, it is necessary to clarify the physiological status of the fish following exposure.
- The manuscript text makes it difficult to ascertain the number of biological replicates used for the transcriptomic analysis. An examination of the data presented in Table 1 indicates that there were five replicates for each group. This fact should be clearly stated in the "Materials and Methods" section.
- The article is accompanied by supplementary materials; however, these materials are not referenced within the text of the article. These elements must be incorporated into the study.
- The captions for the diagrams and graphs in Figure 3 are illegible. The authors should consider enhancing the quality of the graphs or alternatively, they may consider moving the original illustrations to the supplement section.
- It is recommended that the authors revise the text in the "Discussion" section, dividing the semantic blocks into separate paragraphs. This will enhance the reader's perception of the information.
- When describing the morphological features of atresia, the authors likely omitted a reference to a literature source (line 351).

Author Response

Specific comments:

Comment 1: The authors should specify the amount of biomaterial (e.g., 5–10 g) collected for research (line 128).

Reply: Thank you very much, I agree with you. Please check it on page 3. Line 136.

Comment 2: The “Materials and Methods” section states that during the experimental exposure, the fish were kept in pools with a diameter of 1 m (line 133). How appropriate is it to keep specimens measuring 111–120 cm (line 122) in such containers? Additionally, the authors should specify the depth of the pools.

Reply: Thank you for your reminder. We add it to the paper. We made mistakes, it should be 10m for diameter. Please check it on page 3. Line 139.

Comment 3: The authors must provide a thorough justification for the selection of ammonium concentrations employed in the study. Additionally, the researchers should specify which inorganic salt or other reagent was utilized in order to obtain the stated concentrations. It is my estimation that these concentrations approximate lethal levels; therefore, it is necessary to clarify the physiological status of the fish following exposure.

Reply: Thank you for your suggestion, we describe it on the paper, please check it on page3. Line 141 to 144

Comment 4: The manuscript text makes it difficult to ascertain the number of biological replicates used for the transcriptomic analysis. An examination of the data presented in Table 1 indicates that there were five replicates for each group. This fact should be clearly stated in the "Materials and Methods" section.

Reply: Thank you for your suggestion , we have modified it and please check it on page 4, line 176

Comment 5: The article is accompanied by supplementary materials; however, these materials are not referenced within the text of the article. These elements must be incorporated into the study.

Reply: Thank you for your suggestion, but we marked it before, please check it on page 5, line 215; Page 6, Line 237; Page 10 Line314;

Comment 6: The captions for the diagrams and graphs in Figure 3 are illegible. The authors should consider enhancing the quality of the graphs or alternatively, they may consider moving the original illustrations to the supplement section.

Reply: Thank you very much for your suggestion, we remade it. Please check it on page 9, Figure3.

Comment 7: It is recommended that the authors revise the text in the "Discussion" section, dividing the semantic blocks into separate paragraphs. This will enhance the reader's perception of the information.

Reply: Thank you for your suggestion , we made it and please check it on discussion part. From page 12 to page 15.

Comment 8: When describing the morphological features of atresia, the authors likely omitted a reference to a literature source (line 351).

Reply: Thank you for your suggestion. In this part, we described our results and without checking any references.

Reviewer 3 Report

Comments and Suggestions for Authors

Dear Authors, 

The manuscript presents new data and is extremely important for studying the effects of ammonia stress in the mature eggs of the Acipenser baerii. There are several comments, which I think should be considered before publication in the journal. 

In general, in the text:
• There is no data on transcriptome validation using the real-time PCR method.
• There is no information about whether the source data of the article is publicly available (Data Availability Statemen). I could not find any indication about the data generated in this study being submitted to NCBI or another database, nor could I find this information by searching NCBI. This aspect is crucial and MANDATORY to allow in the future retrospective analysis of fastq files. It is the right of the authors to keep the information private until the publication of the manuscript, but please submit them and provide a reviewer’s token in this case. It is crucial for the reproducibility of the analysis. As the most time effective way to publish the data, I would recommend the NCBI GEO database. It allows for the deposition of both raw and processed data (as well as assemblies), processes the data quite rapidly and allows for the generation of a reviewer’s token. On top of that, it would be ideal to also submit the assembly to the TSA/GenBank database so that it is subsequently included in the databases used by the BLAST toolkit.

Title: Ammonia stress induced transcriptional expression changes in 2
mature eggs of the Acipenser baerii - please, add "the" before "mature eggs"

Lines 87-93: "This method is now widely used to investigate the environmental stress response of aquatic animals, such as hypoxic stress on zebrafish (Danio rerio) (Woods & Imam, 2015), different salinity stress on money fish (Scatophagus argus) (Su et al., 2019), ammonia stress on largemouth black bass (Micropterus salmoides) (Zou et al., 2023) and Japanese shrimp (Marsupenaeus japonicus) (Liang et al., 2019), spotted shrimp (Penaeus monodon) (Li et al., 2018), and grass carp (Ctenopharyngodon idellus) (Jin et al., 2017)." - There are numerous papers on this topic (using sequencing), particularly those on environmental stress factors, so citing individual publications, especially those from 2015-2017, seems unnecessary. Please supplement the list with more recent papers or cite review articles.

Line 167: There is no description of how to construct sequencing libraries.

Lines 171, 172, 177, 191: The methodology section lacks references to the programs used in the work, such as FASTQ, Trimmomatic, Trinity, and WGCNA, respectively. The program versions are also not specified (except for Trimomatic (version 0.39))

Lines 169-170: There is no reference to the official Illumina standard operating procedures (SOPs), which were mentioned.

Figure 1. - Resolution on the scale is almost illegible.

Lines 210-212: "This study collected a total of 15 mature egg samples. High-throughput RNA sequencing was performed on 15 samples, and after sequencing was completed, raw data were obtained. The raw data was then evaluated and quality controlled using Fastqc and TrimmaticV0.39." - This data is for the Methods chapter, not the Results chapter.

In Chapters 3.2 and 3.3, you write about 15 samples, but it is better to indicate 5 samples for each experimental variant to avoid confusion.

3. Differential expression analysis - correct to 3.4.

Lines 235-236: Differential expression analysis was performed using DEseq2 software with thresholds of |log₂(Fold change)|≥1 and P<0.05, identifying 18,741 DEGs. - This information is for the Methods chapter.

Line 267: Figure 3 A, B, C – completely unreadable text (GO term).
Figure 3 D, E – poorly readable text (KEGG pathway)

Line 340: Typo: extra capital letter in the word "Light" in the middle of the sentence.

Author Response

Comment 1: There is no data on transcriptome validation using the real-time PCR method.

Reply: Thank you for pointing this out, we agree with this comment. Therefore, we validated for these genes by using RT-qPCR. I will show it on page 12, Figure 5

  • Comment 2: There is no information about whether the source data of the article is publicly available (Data Availability Statemen). I could not find any indication about the data generated in this study being submitted to NCBI or another database, nor could I find this information by searching NCBI. This aspect is crucial and MANDATORY to allow in the future retrospective analysis of fastq files. It is the right of the authors to keep the information private until the publication of the manuscript, but please submit them and provide a reviewer’s token in this case. It is crucial for the reproducibility of the analysis. As the most time effective way to publish the data, I would recommend the NCBI GEO database. It allows for the deposition of both raw and processed data (as well as assemblies), processes the data quite rapidly and allows for the generation of a reviewer’s token. On top of that, it would be ideal to also submit the assembly to the TSA/GenBank database so that it is subsequently included in the databases used by the BLAST toolkit.

Reply: Thank you for pointing this out, we agree with this comment. Therefore, we uploaded the RNA-seq to NCBI, but we need to wait for a few days.

Comment 3 :Title: Ammonia stress induced transcriptional expression changes in 2mature eggs of the Acipenser baerii - please, add "the" before "mature eggs"

Reply: Thank you for pointing this out, we agree with this comment. We add this one. I will show it on page 1 Line 2

Comment 4:Lines 87-93: "This method is now widely used to investigate the environmental stress response of aquatic animals, such as hypoxic stress on zebrafish (Danio rerio) (Woods & Imam, 2015), different salinity stress on money fish (Scatophagus argus) (Su et al., 2019), ammonia stress on largemouth black bass (Micropterus salmoides) (Zou et al., 2023) and Japanese shrimp (Marsupenaeus japonicus) (Liang et al., 2019), spotted shrimp (Penaeus monodon) (Li et al., 2018), and grass carp (Ctenopharyngodon idellus) (Jin et al., 2017)." - There are numerous papers on this topic (using sequencing), particularly those on environmental stress factors, so citing individual publications, especially those from 2015-2017, seems unnecessary. Please supplement the list with more recent papers or cite review articles.

Reply: Thank you very much for your suggestion. We have changed it and listed the newest paper for reference. I will show it on page 2. Line 89 to 93.

Comment 5: Line 167: There is no description of how to construct sequencing libraries.

Reply: Thank you for pointing out, we added this part on the paper. I will show it on page 4 Line 176

Comment 6: Lines 171, 172, 177, 191: The methodology section lacks references to the programs used in the work, such as FASTQ, Trimmomatic, Trinity, and WGCNA, respectively. The program versions are also not specified (except for Trimomatic (version 0.39))

Reply: Thank you for your suggestion, we supplemented it on the paper. I will show it on page 4, Line 177

Comments 7 :Lines 169-170: There is no reference to the official Illumina standard operating procedures (SOPs), which were mentioned.

Reply: Thank you for remind. We made it.

Comment 8: Figure 1. - Resolution on the scale is almost illegible.

Reply: We remade the figure1, please check it on the figures. I will show it on page 5. Figure 1

Comment 9 :Lines 210-212: "This study collected a total of 15 mature egg samples. High-throughput RNA sequencing was performed on 15 samples, and after sequencing was completed, raw data were obtained. The raw data was then evaluated and quality controlled using Fastqc and TrimmaticV0.39." - This data is for the Methods chapter, not the Results chapter.

Reply:Thank you for remind. We deleted it and moved to material and methods. I will show it on Line230-233, Page 6

Comment 10: In Chapters 3.2 and 3.3, you write about 15 samples, but it is better to indicate 5 samples for each experimental variant to avoid confusion.

Reply: Thank you for remind. We described it more clear. I will show it on page 4, Line: 176

Comment 11: Differential expression analysis - correct to 3.4.

Reply: Thank you very much, we modified it.

Comment 12: Lines 235-236: Differential expression analysis was performed using DEseq2 software with thresholds of |log₂(Fold change)|≥1 and P<0.05, identifying 18,741 DEGs. - This information is for the Methods chapter.

Reply:Thank you very much, we modified it.

Comment 13: Line 267: Figure 3 A, B, C – completely unreadable text (GO term).
Figure 3 D, E – poorly readable text (KEGG pathway)

Reply: Thank you very much, we remade it on the figures. Please check it on Figure 3

Line 340: Typo: extra capital letter in the word "Light" in the middle of the sentence.

Reply: Thank you very much, we modified it.

28 Sep 2025 10:46:12

Round 2

Reviewer 2 Report

Comments and Suggestions for Authors

The authors have carefully considered the comments and made all necessary corrections to the manuscript. The article can be published in its current form.

Author Response

Comments: The authors have carefully considered the comments and made all necessary corrections to the manuscript. The article can be published in its current form.

Reply: Thank you very much

Reviewer 3 Report

Comments and Suggestions for Authors

The authors did a great job revising their manuscript, and I’m happy my comments were of use. The authors took in consideration all my comments and addressed them properly.

Minor revision:
Lines 183-190: "2.8. Quantitative PCR protocol" - It would be better to title this chapter "Validation of data via qPCR". This chapter lacks a detailed description. You only list genes in the results (3.7. RNA-seq expression analysis), but they should also be described in the methods chapter. The genes should be written in italics, there is no primers sequence:
Was the RNA purified from background DNA? If so, please specify. Which genes were chosen for transcriptome validation and why? Primer sequences should be provided in the table; the methods should describe how the sequences were selected. Which gene was chosen as the reference gene? Why was this gene chosen? Was the stability of the reference gene assessed (see MIQE guidelines)? How many samples were used for this method (specify this information in Chapter 2.8)? In what application was the 2-△△CT method used? On what equipment?

I have also one comment at the Editor’ discretion. I am confused about when exactly the authors will publish the raw data in NCBI, whether there will be any mention of the access number or link in the article. If you do not include the access number in the article to NCBI at this stage, then it will probably be difficult to do so later. Below I again provide my question and the authors’ comment on it. Perhaps the journal may publish the accession number later?

Comment 2: There is no information about whether the source data of the article is publicly available (Data Availability Statemen). I could not find any indication about the data generated in this study being submitted to NCBI or another database, nor could I find this information by searching NCBI. This aspect is crucial and MANDATORY to allow in the future retrospective analysis of fastq files. It is the right of the authors to keep the information private until the publication of the manuscript, but please submit them and provide a reviewer’s token in this case. It is crucial for the reproducibility of the analysis. As the most time effective way to publish the data, I would recommend the NCBI GEO database. It allows for the deposition of both raw and processed data (as well as assemblies), processes the data quite rapidly and allows for the generation of a reviewer’s token. On top of that, it would be ideal to also submit the assembly to the TSA/GenBank database so that it is subsequently included in the databases used by the BLAST toolkit.

Reply: Thank you for pointing this out, we agree with this comment. Therefore, we uploaded the RNA-seq to NCBI, but we need to wait for a few days.

Author Response

Comments: Lines 183-190: "2.8. Quantitative PCR protocol" - It would be better to title this chapter "Validation of data via qPCR". 

Reply: I agree with you, and I modified it. Please check it on page 4 line 182.

Comments: You only list genes in the results (3.7. RNA-seq expression analysis), but they should also be described in the methods chapter. The genes should be written in italics, there is no primers sequence.

Reply: Thank you very much, I modified methods of RT-qPCR validation, please check it on page 4 line 182-199, for the gene name, I revised it in Italics. please check it in page 11, line 389

Comments: Was the RNA purified from background DNA? If so, please specify. Which genes were chosen for transcriptome validation and why? Primer sequences should be provided in the table; the methods should describe how the sequences were selected. Which gene was chosen as the reference gene? Why was this gene chosen? Was the stability of the reference gene assessed (see MIQE guidelines)? How many samples were used for this method (specify this information in Chapter 2.8)? In what application was the 2-△△CT method used? On what equipment?

Reply: Thank you very much. I modified the methods of RT-qPCR validation. Please check it on page 4 line 182-199. The primer table was shown in Table S2.

Comments: I have also one comment at the Editor’ discretion. I am confused about when exactly the authors will publish the raw data in NCBI, whether there will be any mention of the access number or link in the article. If you do not include the access number in the article to NCBI at this stage, then it will probably be difficult to do so later. Below I again provide my question and the authors’ comment on it. Perhaps the journal may publish the accession number later?

Reply: I apologized for that. The process of RNA sequencing was completed in April. It has been so long that we cannot find the original data yet. Therefore, we contacted the sequencing corporation to release the link for download again, and we will submit the data to NCBI within this week.
